# On the Out-of-distribution Generalization of Probabilistic Image Modelling

**Mingtian Zhang** [1,2]*       **Andi Zhang** [2,3]*       **Steven McDonagh** [2]

[1]AI Center, University College London, [2]Huawei Noah's Ark Lab,
[3] Department of Computer Science and Technology, University of Cambridge
mingtian.zhang.17@ucl.ac.uk    az381@cam.ac.uk    steven.mcdonagh@huawei.com

## Abstract

Out-of-distribution (OOD) detection and lossless compression constitute two problems that can be solved by the training of probabilistic models on a first dataset with subsequent likelihood evaluation on a second dataset, where data distributions differ. By defining the generalization of probabilistic models in terms of likelihood we show that, in the case of image models, the OOD generalization ability is dominated by local features. This motivates our proposal of a *Local Autoregressive* model that exclusively models local image features towards improving OOD performance. We apply the proposed model to OOD detection tasks and achieve state-of-the-art unsupervised OOD detection performance *without* the introduction of additional data. Additionally, we employ our model to build a new lossless image compressor: NeLLoC (Neural Local Lossless Compressor) and report state-of-the-art compression rates and model size.

## 1   Introduction

Probabilistic modeling has achieved great success in the modeling of images. Likelihood based models, *e.g.* Variational Auto-Encoders (VAE) [22], Flow [21], Pixel CNN [49, 41] are shown to successfully generate high quality images, in addition to estimating underlying densities.

The goal of probabilistic modelling is to use a model $p_\theta$ to approximate the unknown data distribution $p_d$ using the training data $\{x_1, \dots, x_N\} \sim p_d$. A common method to learn parameters $\theta$ is to minimize some divergence between $p_d$ and $p_\theta$, for example, a popular choice is the KL divergence

$$\text{KL}(p_d || p_\theta) = -\text{H}(p_d) - \int \log p_\theta(x) p_d(x) dx, \tag{1}$$

where the entropy of the data distribution is a constant. Since we only have access to finite $p_d$ data samples $x_1, \dots, x_N$, the second term is typically approximated using a Monte Carlo estimation

$$\text{KL}(p_d || p_\theta) \approx -\frac{1}{N} \sum_{n=1}^{N} \log p_\theta(x_n) - const.. \tag{2}$$

Minimizing the KL divergence is therefore equivalent to Maximum Likelihood Estimation. A typical evaluation criterion for the learned models is the test likelihood $\frac{1}{M} \sum_{m=1}^{M} \log p_\theta(x_m)$, with test set $\{x_1, \dots, x_M\} \sim p_d$. We refer to this evaluation as *in-distribution (ID) generalization*, since both training and test data are sampled from the same distribution $p_d$. However, in this work, we are interested in *out-of-distribution (OOD) generalization* such that the test data are drawn from $p_o$, where $p_o \neq p_d$. To motivate our study of this topic, we firstly introduce two applications: lossless compression and OOD detection, that both can make use of the OOD generalization property.

---

*Co-first author, the work was done during an internship in Huawei Noah's Ark Lab.

35th Conference on Neural Information Processing Systems (NeurIPS 2021).

## 1.1 Model-based Lossless Compression

Lossless compression has a strong connection to probabilistic models [33]. Let $\{x_1, \ldots, x_M\}$ be test data to compress, where $x_m$ is sampled from some underlying distribution with PMF $p_d$. If $p_d$ is known, we can design a compression scheme to compress each data $x_m$ to a bit string with length approximately $-\log_2 p_d(x_m)^2$. As $M \to \infty$, the averaged compression length will approach the entropy of the distribution $p_d$, that is; $\frac{1}{M}\sum_{m=1}^{M} -\log_2 p_d(x_m) \to \mathrm{H}(p_d)$ where $\mathrm{H}(\cdot)$ denotes entropy. In this case, the compression length is *optimal* under Shannon's source coding theorem [45], *i.e.* we cannot find another compression scheme with compression length less than $\mathrm{H}(p_d)$. In practice, however, the true data distribution $p_d$ is unknown and we must use a model $p_\theta \approx p_d$ to build the lossless compressor. Recent work successfully apply deep generative models such as VAEs [47, 48, 23], Flow [19, 3] to conduct lossless compression. We note that the underlying models are designed to focus on test data that follows the same distribution as the training data, resulting in test compression rates that depend on the ID generalization ability of the model.

However, in practical compression applications, the distribution of the incoming test data is unknown, and is usually different from the training distribution $p_d$: $\mathcal{X}_{test}^o = \{x_1', \ldots, x_M'\}$ where $x' \sim p_o \neq p_d$. To achieve good compression performance, a lossless compressor model should still be able to assign *high* likelihood for these OOD data. This practical consideration motivates **encouragement of the OOD generalization ability** of the model.

Empirical results [48, 19] have shown that we can still use model $p_\theta$ (trained to approximate $p_d$) in order to compress test data $\mathcal{X}_{test}^o$ with reasonable compression rates. However, the phenomenon that these models can generalize to OOD data lacks intuition and key components, affecting this generalization ability, remain underexplored. Consideration of recent advances in likelihood-based OOD detection next allows us to further investigate these questions and lead to our proposal of a new model that can encourage OOD generalization.

## 1.2 Likelihood-based OOD Detection

Given a set of unlabeled data, sampled from $p_d$, and a test data $x'$ then the goal of OOD detection is to distinguish whether or not $x'$ originates from $p_d$. A natural approach [5] involves fitting a model $p_\theta$ to approximate $p_d$ and treat sample $x'$ as in-distribution if its (log) likelihood is larger than a threshold; $\log p_\theta(x') > \epsilon$. Therefore, a good OOD detector model should assign *low* likelihood to the OOD data. In contrast to lossless compression, this motivates **discouragement of the OOD generalization ability of the model.**

Surprisingly, recent work [36] report results showing that popular deep generative models, including *e.g.* VAE [22], Flow [21] and PixelCNN [41], can assign OOD data *higher* density values than in-distribution samples, where such OOD data may contain differing semantics, *c.f.* the samples used for maximum likelihood training. We demonstrate this phenomenon using PixelCNN models, trained on Fashion MNIST (CIFAR10) and tested using MNIST (SVHN). Figure 1 provides histograms of model evaluation using negative bits-per-dimension (BPD), that is; the $\log_2$ likelihood normalized by data sample dimension (larger negative BPD corresponds to larger likelihood). We corroborate previous work and observe that tested models assign higher likelihood to the OOD data, in both cases. This counter intuitive phenomenon suggests that likelihood-based approaches may not make for good OOD image detection criterion, yet encouragingly also illustrates that a probabilistic model, trained using one dataset, may be employed to compress data originating from a different distribution with a potentially higher compression rate. This intuition builds a connection between OOD detection and lossless compression. Inspired by this link, we next investigate the underlying latent causes of image model generalizability, towards improving both lossless compression and OOD detection.

## 2 OOD Generalizations of Probabilistic Image Models

Previous work studies the potential causes of the surprising OOD detection phenomenon: OOD data may have higher model likelihood than ID data. For example, [37] used a typical set to reason about the source of the effect, while the work of [40] argues that likelihoods are significantly affected by

---

[2]For a compression method like Arithmetic Coding [50], the message length is always within two bits of $-\log_2 p_d(x_m)$ [33], also see Section 4.1.

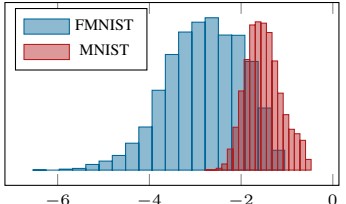 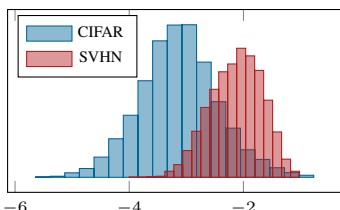

Figure 1: Left: log likelihood of FashionMNIST, MNIST test data using a full PixelCNN model, trained on FashionMNIST training set. Right: log likelihood of CIFAR10, SVHN test data using a PixelCNN model, trained on CIFAR10 training set. The $x$-axis indicates the value of the log-likelihood (negative BPD), the $y$-axis provides data sample counts.

image background statistics or by the size and smoothness of the background [26]. In this work, we alternatively consider a recent hypothesis proposed by [43] (also implicitly discussed in [24] for a flow-based model): *low-level local features, learned by (CNN-based) probabilistic models, are common to all images and dominate the likelihood.* From the perspective of OOD generalization, this allows formulation of the following related conjectures:

- Models **can** generalize to OOD images as local features are shared between image distributions.
- Models can generalize **well** to OOD images since local features dominate the likelihood.

In the work of [43], the authors investigated their original hypothesis by studying the differences between individual pixel values and neighbourhood mean values and additionally considered the correlation between models trained on small image patches and trained, alternatively, on full images. To further investigate this hypothesis, we rather propose to *directly model the in-distribution dataset, using only local feature information.* If the hypothesis is true, then the proposed *local* model alone should generalize well to OOD images. By contrasting such an approach with a standard *full* model, that considers both local and non-local features, we are also able to study the contribution that local features make to the full model likelihood. We first discuss how to build a local model for the image distribution and then use the proposed model to study generalization on OOD datasets.

## 2.1 Local Model Design

Autoregressive models have proven popular for modeling image datasets and common instantiations include PixelCNN [49, 41], PixelRNN [49] and Transformer based models [7]. Assuming data dimension $D$, the Autoregressive model $p_f(x)$ can be written as

$$p_f(x) = p(x_1) \prod_{d=2}^{D} p(x_d|x_1, \ldots, x_{d-1}), \tag{3}$$

informally we refer to this type of model as a "full model" since it can capture all dependencies between each dimension (pixel). Similarly, we can define a local autoregressive model $p_l(x)$ where pixel $x_{ij}$, at image row $i$ column $j$, depends on previous pixels with fixed horizon $h$:

$$p_l(x) = \prod_{ij} p(x_{ij}|x_{[i-h:i-1,j-h:j+h]}, x_{[i,j-h:j-1]}), \tag{4}$$

with zero-padding used in cases where $i$ or $j$ are smaller than $h$. Figure 2b illustrates the resulting local autoregressive model dependency relationships. We implement this model using a masked CNN [49], with kernel size $k=2\times h+1$ in our first network layer, to mask out future pixel dependency. A full PixelCNN model would then proceed to stack multiple masked CNN layers, where increasing kernel depth affords receptive field increases. In contrast, we employ masked CNN with $1\times1$ convolutions in subsequent layers. Such $1\times1$ convolutions can model the correlation between channels, as in [21], and additionally prevent our local model obtaining information from pixels outwith the local feature region, defined by $h$. Pixel dependencies are therefore defined solely using the kernel size of the first masked CNN layer, allowing for easy control over model local feature size. We note that the proposed local autoregressive model can also be implemented using alternative backbones *e.g.* Pixel RNN [49] or Transformers [7]. We plot local model samples in Figure 4. Unlike full autoregressive models [49, 41], which can be used to generate semantically coherent samples, we find the samples from the local model are locally consistent yet have no clear semantic meaning.

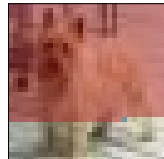
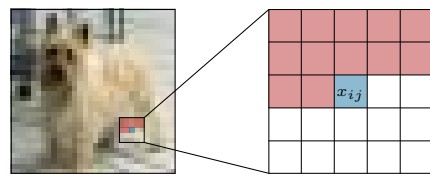

|              (a) Full Autoregressive Model              |              (b) Local Autoregressive Model              |

Figure 2: (a) full autoregressive model pixel dependencies; the distribution of the current pixel (blue) depends on all *previous* pixels (red); (b) local autoregressive model dependencies, with $h = 2$. The distribution of $x_{ij}$ (blue) depends on only the pixels in a local region (red).

## 2.2 Local Model Generalization

To investigate the generalization ability of our local autoregressive model, we fit the model to Fashion MNIST (grayscale) and CIFAR10 (color) training datasets and test using in-distribution (ID) images (respective dataset test images) and additional out-of-distribution (OOD) datasets: MNIST, KMNIST (grayscale) and SVHN, CelebA[3] (color). Both models use the discretized mixture of logistic distributions [41] with 10 mixtures for the predictive distribution and a ResNet architecture [49, 14]. We use a local horizon length $h=3$ (kernel size $k=7$) for both grayscale and color image data. We compare our local autoregressive model to a standard full autoregressive model (*i.e.* a standard PixelCNN), with additional network architecture and training details found in Appendix A. Tables 1, 2

Table 1: Test BPD (Trained on Fashion MNIST)

| Test Dataset        | Full | Local |
|---------------------|------|-------|
| Fashion MNIST (ID)  | 2.78 | 2.89  |
| MNIST (OOD)         | 1.50 | 1.49  |
| KMNIST (OOD)        | 2.48 | 2.44  |

Table 2: Test BPD (Trained on CIFAR10)

| Test Dataset   | Full | Local |
|----------------|------|-------|
| CIFAR10 (ID)   | 3.12 | 3.25  |
| SVHN (OOD)     | 2.13 | 2.13  |
| CelebA (OOD)   | 3.33 | 3.35  |

report comparisons in terms of BPD (where lower values entail higher likelihood) for Fasion MNIST and CIFAR10, respectively. We observe that for in-distribution (ID) data, the full model has better generalization ability *c.f.* the local model (0.11 and 0.13 BPD, respectively). This is unsurprising as training and test data originate from the same distribution; both local and non-local features, as learned by the full model, help ID generalization. For OOD data, we observe that the local model has generalization ability similar to the full model, exhibiting very small empirical gaps (only $\approx 0.02$ BPD on average), showing that the local model alone can generalize well to OOD distributions. We thus verify the hypothesis considered at the start of Section 2.

For simple datasets containing gray-scale images, the PixelCNN model is flexible enough to capture both local and global features. We notice that, in Table 1, our local model exhibits even better OOD generalization than the full model. This drives us to further study the role of non-local features for generalization. When the local horizon size increases, the model will be able to learn features with greater non-locality. We thus vary the local horizon size to study generalization ability under this property, see Table 3. We find the model has poor generalization performance when local features are too small and increasing the horizon size helps ID generalization but decreases the OOD generalization. A consistent phenomenon is observed when considering color images, see Appendix B.1. We can thus conclude: **non-local features are not shared between images distributions, overfitting to non-local features will hurt generalization**.

These hypotheses indicate two opposing strategies for the considered tasks:

*OOD detection:* distributions are distinguished by dataset-unique features, thus building **non-local** models, able to discount common local features, improves the detector distinguishability power.

Table 3: Generalization of local model with different horizon sizes. The model is trained on FashionMNIST dataset.

| Method       | h=1  | h=2  | h=3  | h=4  | h=5  |
|--------------|------|------|------|------|------|
| (ID) Fashion | 3.17 | 2.93 | 2.89 | 2.88 | 2.88 |
| (OOD) MNIST  | 1.54 | **1.48** | 1.49 | 1.50 | 1.51 |
| (OOD) KMNIST | 2.54 | **2.43** | 2.44 | 2.46 | 2.47 |

---

[3]We down-sample the original CelebA to $32\times32\times3$, see Appendix A.2 for details.

*Lossless compression:* OOD generalization is possible due to the sharing of local features between distributions. Employing only a local model can encourage OOD generalization, by preventing the model from over-fitting to dataset-unique features, specific to the training distribution.

Sections 3 and 4 will further demonstrate how contrasting modeling strategies can benefit these tasks.

## 3 OOD Detection with Non-Local Model

As was discussed in Section 1.2, local image features are observed to be largely common across the real-world image distribution and can be treated as a domain-prior. Therefore, in order to detect whether or not an image is out-of-distribution, we can stipulate a non-local model able to discount local features of the image distribution; denoted here $p_{nl}(x)$. It is however not easy to build such a non-local model directly, since the concept of "non-local" lacks a mathematically rigorous definition. However, we propose that a non-local model can be considered to be the complement of a local model, from a respective full model. In the following section, we therefore propose to use a product of experts to indirectly define $p_{nl}(x)$, and demonstrate how this may be used for OOD detection.

### 3.1 Product of Experts and Non-Local Model

As demonstrated in Section 2.2, the full model $p_f(x)$ and the local model $p_l(x)$ can be easily built for the image distribution, *e.g.* a full autoregressive model and a local autoregressive model. We further assume the full model allows the following decomposition:

$$p_f(x) = \frac{p_l(x)p_{nl}(x)}{Z}, \tag{5}$$

where $Z = \int p_l(x)p_{nl}(x)dx$ is the normalizing constant. This formulation can also be thought of as a product of experts (PoE) model [17] with two experts; $p_l$ (local expert) and $p_{nl}$ (non-local expert). An interesting property of the PoE model is that if a data sample $x'$ has high full model probability $p_f(x')$, it should possess high probability mass in *each* of the expert components[4]. Therefore, the PoE model assumption is consistent with our image modelling intuition: a valid image requires both valid local features (*e.g.* stroke, texture, local consistency) and valid non-local features (semantics).

By our model assumption, the density function of the non-local model can be formally defined and is proportional to the likelihood ratio:

$$p_{nl}(x) \propto \frac{p_f(x)}{p_l(x)} \equiv \hat{p}_{nl}(x), \tag{6}$$

where $\hat{p}_{nl}(x)$ denotes the unnormalized density. For the OOD detection task, we require only $\hat{p}_{nl}(x)$ in order to provide a score classifying whether or not test data $x$ is OOD and therefore *do not require to estimate the normalization constant $Z$*. We also note that as we increase the local horizon length for $p_l$, the local model will converge to a full model $p_l \to p_f$, and $\hat{p}_{nl}(x) = 1$ becomes a constant and inadequate for OOD detection. This further suggests the importance of using a local model. Figure 3 shows histograms of $\hat{p}_{nl}(\cdot)$ for both ID and OOD test datasets. We observe that the majority of ID test data obtains higher likelihood than OOD data, illustrating the effectiveness of non-local models.

### 3.2 Connections to Related Methods

We highlight that the score $\hat{p}_{nl}(x)$ that we use to conduct OOD detection allows a principled likelihood interpretation: **the unnormalized likelihood of a non-local model.** We believe this to be the first time that the likelihood of a non-local model is considered in the literature. However, other likelihood ratio variants have been previously explored for OOD detection. We briefly discuss related work and highlight where our method differs from relevant literature.

In [40], it is assumed that each data sample $x$ can be factorized as $x = \{x_b, x_s\}$, where $x_b$ is a background component, characterized by population level background statistics: $p_b$. Further, $x_s$

---

[4]This PoE property differs from a Mixture of Experts (MoE) model that linearly combines experts. If data $x'$ has high probability in the local model (*e.g.* $p_l(x') = 0.9$) but low probability in the non-local model (*e.g.* $p_{nl}(x') = 0.1$), the probability in the full PoE model $p_f(x') \propto 0.9 * 0.1$ is also *small*. On the contrary, if we assume $p_f$ is a MoE: $p_f = \frac{1}{2}p_l + \frac{1}{2}p_{nl}$, then $p_l(x') = 0.9$ and $p_{nl}(x') = 0.1$ results in a *high* full MoE model $p_f(x')$ value *i.e.* $p_f(x') = 0.5 * 0.9 + 0.5 * 0.1 = 0.5$. We refer to [17] for additional PoE model details.

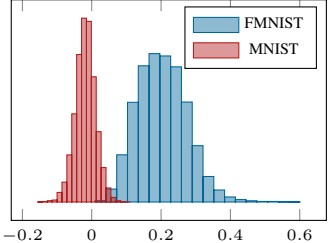 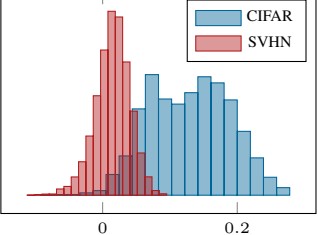 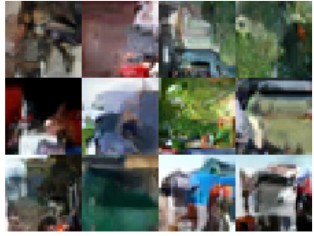

Figure 3: Unnormalized log likelihood of the non-local model $\hat{p}_{nl}(x)=\frac{p_f(x)}{p_l(x)}$ for both ID test datasets (FashionMNIST, CIFAR10) and OOD datasets (MNIST, SVHN). ID test datasets obtain significantly higher likelihoods, on average, in each case.

Figure 4: Samples from the local autoregressive model, see Appendix B.2 for details.

then constitutes a semantic component, characterized by patterns belonging specifically to the in-distribution data: $p_s$. A full model, fitted on the original data, (*e.g.* Flow) can then be factorized as $p_f(x) = p_f(x_b, x_s) = p_b(x_b)p_s(x_s)$ and the semantic model can correspondingly be defined as a ratio: $p_s(x) = \frac{p_f(x)}{p_b(x)}$, where $p_f(x)$ is a full model. In order to estimate $p_b(x_b)$, the authors of [40] design a perturbation scheme and construct samples from $p_b(x_b)$ by adding random perturbation to the input data. A further full generative model is then fitted to the samples towards estimating $p_b$. In our method, both $p_l(x)$ and $p_{nl}(x)$ constitute distributions of the same sample space (that of the original image $x$) whereas $p_s$ and $p_b$ in [40] form distributions in different sample spaces (that of $x_s$ and $x_b$, respectively). Additionally, in comparison with our local and non-local experts factorization of the model distribution, their decomposition of an image into 'background' and 'semantic' parts may not be suitable for all image content and the construction of samples from $p_b(x_b)$ (adding random perturbation) lacks principled explanation. In [44, 43], the score for OOD detection is defined as $s(x) = \frac{p_f(x)}{p_c(x)}$ where $p_f$ is a full model and $p_c$ is a complexity measure containing an image domain prior. In practice, $p_c$ is estimated using a traditional compressor (*e.g.* PNG or FLIF), or a model trained on an additional, larger dataset in an attempt to capture general domain information [43]. In comparison, our method does not require the introduction of new datasets and our explicit local feature model, $p_l$, can be considered more transparent than PNG of FLIF. Additionally, our likelihood ratio can be explained as the *(unnormalized) likelihood of the non-local model* for the in-distribution dataset, whereas the score described by [43, 44] does not offer a likelihood interpretation. In Section 4, we discuss how the proposed local model may be utilized to build a lossless compressor, further highlighting the connection between our OOD detection framework and traditional lossless compressors (*e.g.* PNG or FLIF). In Table 6, we report experimental results showing that a lossless compressor based on our model significantly improves compression rates *c.f.* PNG and FLIF, further suggesting the benefits of the introduced OOD detection method.

### 3.3 Experiments

We conduct OOD detection experiments using four different dataset-pairs that are considered challenging [36]: Fashion MNIST (ID) *vs.* MNSIT (OOD); Fashion MNIST (ID) *vs.* OMNIGLOT (OOD); CIFAR10 (ID) *vs.* SVHN (OOD); CIFAR10 (ID) *vs.* CelebA (OOD). We actively select not to include dataset pairs such as CIFAR10 *vs.* CIFAR100 or CIFAR10 *vs.* ImageNet since these contain duplicate classes and cannot be treated as strictly disjoint (or OOD) datasets [44]. Additional experimental details are provided in Appendix A. In Table 4 we report the 'area under the receiver operating characteristic curve' (AUROC), a common measure for the OOD detection task [15]. We compare against methods, some of which require additional label information[5], or datasets. Our method achieves state-of-the-art performance in most cases without requiring additional information. We observed that the model-ensemble methods, WAIC [8] and MSMA [34] can achieve higher AUROC in the experiments involving color images yet are significantly outperformed by our approach in the case of grayscale data. We thus evidence that our simple method is consistently more reliable than alternative approaches and that our score function allows a principled likelihood interpretation.

---

[5]In principle methods that require labels correspond to *classification*, task-dependent OOD detection, which may be considered fundamentally different from task-independent OOD detection (with access to only image space information), see [1] for details. We compare against both classes of method, for completeness.

Table 4: OOD detection comparisons (AUROC). Higher values indicate better performance, results are rounded to three decimal places. Results are reported in each case directly using the original references except in the cases of ODIN [40, 29] and VIB [8]. Results for Typicality test are from [44], corresponding to batches of two samples of the same type. (a) The Mahalanobis method requires knowledge of the validation data (OOD distribution). (b) A full PixelCNN (see Appendix A) is trained on the ID dataset and its likelihood evaluations are then used to calculate AUROC.

| ID dataset: | FashionMNIST | | CIFAR10 | |
| OOD dataset: | MNIST | Omniglot | SVHN | CelebA |
| --- | --- | --- | --- | --- |
| Using Labels | | | | |
| ODIN [31] | 0.697 | - | 0.966 | - |
| VIB [2] | 0.941 | 0.943 | 0.528 | 0.735 |
| Mahalanobis[a] [16] | 0.986 | - | 0.991 | - |
| Gram-Matrix [42] | - | - | 0.995 | - |
| Using Additional Datasets | | | | |
| Outlier Exposure [16] | - | - | 0.758 | 0.615 |
| Glow diff to Tiny-Glow [43] | - | - | 0.939 | - |
| PCNN diff to Tiny-PCNN [43] | - | - | 0.944 | - |
| Not Using Additional Information | | | | |
| WAIC (model ensemble) [8] | 0.766 | 0.796 | **1.000** | - |
| Glow diff to PNG [43] | - | - | 0.754 | - |
| PixelCNN diff to PNG [43] | - | - | 0.823 | - |
| Likelihood Ratio in [40] | 0.997 | - | 0.912 | - |
| MSMA KD Tree [34] | 0.693 | - | 0.991 | - |
| S using Glow and FLIF [44] | 0.998 | **1.000** | 0.950 | 0.736 |
| S using PCNN and FLIF [44] | 0.967 | **1.000** | 0.929 | 0.535 |
| Full PixelCNN likelihood[b] | 0.074 | 0.361 | 0.113 | 0.602 |
| **Our method** | **1.000** | **1.000** | 0.969 | **0.949** |

# 4 Lossless Compression with Local Model

Recent deep generative model based compressors [47, 3, 23, 18] are designed under the assumption that data to be compressed originates from the same distribution (source) as model training data. However, in practical scenarios, test images may come from a diverse set of categories or domains and training images may be comparatively limited [33]. Obtaining a single method capable of offering strong compresson performance on data from different sources remains an open problem and related study involves consideration of "universal" compression methods [33]. Based on our previous intuitions relating to generalization ability; to build such a "universal" compressor in the image domain, we believe a promising route involves leveraging models that only depend on common local features, shared between differing image distributions. We thus propose a new "universal" lossless image compressor: NeLLoC (Neural Local Lossless Compressor), built upon the proposed local autoregressive model and the concept of Arithmetic Coding [50]. In comparison with alternative recent deep generative model based compressors, we find that NeLLoC has competitive compression rates on a diverse set of data, yet requires significantly smaller model sizes which in turn reduces storage space and computation costs. We further note that due to our design choices, and in contrast to alternatives, NeLLoC can compress images of *arbitrary* size.

In the remaining parts of this section, we firstly discuss NeLLoC implementation and then provide further details on the most important resulting properties of the method.

## 4.1 Implementation of NeLLoC

Our NeLLoC implementation uses the same network backbone as that of our OOD detection experiment (Section 3): a Masked CNN with kernel size $k = 2 \times h + 1$ ($h$ is the horizon size) in the first layer and followed by several residual blocks with $1 \times 1$ convolution, see Appendix A for the network

architecture and training details. To realize the predictive distribution for each pixel, we propose to use a discretized Logistic-Uniform mixture distribution, which we now introduce.

**Discretized Logistic-Uniform Mixture Distribution** The discretized logistic mixture distribution, proposed by [41], has shown promising results for the task of modeling color images. In order to ensure numerical stability, the original implementation can provide only an (accurate) approximation and therefore cannot be used for our task of lossless compression, which requires exact numerical evaluation. We therefore propose to use the discretized Logistic-Uniform Mixture distribution, which mixes the original discretized logistic mixture distribution with a discrete uniform distribution:

$$x \sim (1 - \alpha) \left( \sum_{i=1}^{K} \pi_i \text{Logistic}(\mu_i, s_i) \right) + \alpha U(0, \ldots, 255), \tag{7}$$

where $U(0, \ldots, 255)$ is the discrete uniform distribution over the support $\{0, \ldots, 255\}$. The proposed mixture distribution can explicitly avoid numerical issues and its PMF and CDF can be easily evaluated without requiring approximation. We can then use this CDF evaluation in relation to Arithmetic Coding. In practice; we set $\alpha = 10^{-4}$ to balance numerical stability and model flexibility. We use $K = 10$ (mixture components) for all models in the compression task.

**Arithmetic Coding** Arithmetic coding (AC) [50] is a form of entropy encoding which utilizes a (discrete) probabilistic model $p(\cdot)$ to map a datapoint $x$ to an interval $[0, 1]$. One fraction, lying in the interval, can then be used to represent the data uniquely. If we convert the fraction into a large message stream, the length of the message is always within two bits of $-\log_2 p(x)$ [33]. Our vanilla NeLLoC implementation makes use of the decimal version of AC. In principle, NeLLoC can also be combined with an Asymmetric Numeral System (ANS) [11], which gives faster coding time yet sacrifices compression rate. The work of [13] further proposed interleaved-ANS (iANS), which enables coding a batch of data simultaneously. In Appendix C, we report comparison of NeLLoC-AC, NeLLoC-ANS and NeLLoC-iANS in terms of compression rate and run time[6].

## 4.2 Properties of NeLLoC

**Universal Image Compressor** As discussed previously, the motivation for designing NeLLoC is to realize an image compressor that is applicable (generalizable) to images originating from differing distributions. Towards this, NeLLoC conducts compression depending on local features which are shown to constitute a domain prior for all images. In addition to universality properties, we next discuss other important aspects towards making NeLLoC practical when considering real applications.

**Arbitrary Image Size** Common generative models, *e.g.* VAE or Flow, can only model image distributions with fixed dimension. Therefore, lossless compressors based on such models [47, 19, 3, 18, 23] can only compress images of fixed size. Recently, HiLLoC [48] explore fully convolutional networks, capable of accommodating variable size input images, yet still requires even height and width[7]. L3C [35], based on a multi-scale autoencoder, can compress large images yet also requires height and width to be even. NeLLoC is able to compress images with arbitrary size based on an alternative and simple intuition: *we only model the conditional distribution, based on local neighbouring pixels*. We thus do not model the distribution of the entire image and can therefore, in contrast to HiLLoC and L3C, compress arbitrary image sizes without padding requirements.

To validate method properties, we compare the compression performance of NeLLoC with both traditional image compressors and recently proposed generative model based compressors. We train NeLLoC with horizon length $h = 3$ on two (training) datasets: CIFAR10 ($32 \times 32$) and ImageNet32 ($32 \times 32$) and test on the previously introduced test sets, including both ID and OOD data. We also test on ImageNet64 with size $64 \times 64$ and a full ImageNet[8] test set (with average size $500 \times 374$). Table 6 provides details of the comparison. We find NeLLoC achieves better BPD in a majority of cases. Exceptions include LBB [18] having better ID generalization for CIFAR and HiLLoC [48] having better OOD generalization in the full ImageNet, when the model is trained on ImageNet32.

**Small Model Size** In comparison with traditional codecs such as PNG or FLIF, one major limitation of current deep generative model based compressors is that they require generation and storage of

---

[6]We provide practical implementations of NeLLoC with different coders and pre-trained models at `https://github.com/zmtomorrow/NeLLoC`.

[7]For images with odd height or width, padding is required.

[8]Images with height or width greater than 1000 are removed, resulting in a total of 49032 test images.

models of very large size. For example, HiLLoC [48] contains 24 stochastic hidden layers, resulting in a capacity and parameter size of 156 MegaBytes (MB) using 32-bit floating point model weights. This poses practical challenges relating to both storage and transmission of such models to real, often resource-limited, edge-devices. Since NeLLoC only models the local region, a small network: three Residual blocks with $1 \times 1$ convolutions (except the first layer) is enough to achieve state-of-the-art compression performance. Accordingly the parameter size requirement is only 2.75 MB. We also investigate the compression task under NeLLoC with 1 and 0 residual block, which have parameter size of 1.34 and 0.49 MB respectively, and yet still observe respectable performance. We report model size comparisons in Table 6. In principle, NeLLoC can be also combined with other resource saving techniques such as binary network weights to realize the generative model components. This would further reduce model size [4], which we consider a promising line of future investigation.

**Computation Cost and Speed** The computational complexity and speed for a neural based lossless compressor depends on two stages: (1) Inference stage: in order to compress or decompress the pixel $x_d$, a predictive distribution of $x_d$ needs to be generated by the probabilistic models; (2) Coding stage: uses the pixel value $x_d$ and its distribution to generate the binary code representing $x_d$ (encoding) or uses the predictive distribution and the binary code to recover $x_d$ (decoding). The computation cost and speed of the second stage heavily depends on the implementation (*e.g.* programming language) and the choice of coding algorithms, see Appendix C for a discussion about how different coders trade off between speed and accuracy. We here only discuss the computational cost pertaining to the first inference stage, which is affected by two factors: computational complexity and parallelizability.

*1. Computational complexity:* given an image with size $N \times N$, the full autoregressive model distribution of pixel $x_{ij}$ depends on all previews pixels $p(x_{ij}|x_{[1:i,1:j]})$ and the computational cost of conditional distribution calculation scales as $\mathcal{O}(N^2)$. For latent variable models (*e.g.* [48, 47]) or flow models (*e.g.* [3]) the predictive distribution of $x_i$ depends on all other pixels and therefore also scales with $\mathcal{O}(N^2)$. In direct contrast, our local autoregressive model only depends on local regions with horizon $h$, so computation of $p(x_{ij}|x_{[i-h:i-1,j-h:j+h]}, x_{[i,j-h:j-1]})$ scales with only $\mathcal{O}(h^2)$ and typically $h \ll N$ in practice. This results in significant reduction of computational cost, enabling efficient usage of NeLLoC on CPUs. In Appendix C, we show that NeLLoC, with interleaved ANS coder, is able to compress or decompress the CIFAR10 dataset within 0.1s per image on a CPU. This gives us evidence, for the first time, that computation need not be a limitation of generative lossless compression. Further implementation improvements (*e.g.* C++, faster coders such as tANS [12]), will make NeLLoC a strong candidate to supersede traditional image compression methods.

*2. Parallelizability:* a second key property affecting compression speed is parallelizability. Neural compressors that make use of latent variable models [47, 48] can allow for the distribution of each pixel to be independent (given decompressed latents). Flow models with an isotropic base distribution will also result in appealing decompression times on average. However for a full autoregressive model, the predictive distribution of each pixel must be generated in sequential order and thus the decompression stage cannot be directly parallelized. This results in the decompression stage scaling with image size. In contrast to a full autoregressive model, where sequential decompressing is inevitable, we note that with NeLLoC each pixel only depends on a local region. The fact that multiple pixels exist with non-overlapping local regions allows simultaneous decompression at each step, enabling parallelization. Alternatively, we can split an image into patches and compress each patch individually, thus parallelizing the decompression phase.

Table 5 shows the compression BPD when we split full ImageNet images into patches. BPD slightly increases when we increase the number of patches. Splitting an image into patches assumes each patch is independent, which weakens the predictive performance. This results in a trade-off between compression speed and rate: $\approx 0.1$ BPD overhead can offer $\sim \times 400$ speed up in compression, decompression (assuming infinite compute), potentially proving extremely valuable for practical applications.

Table 5: Parallelization using patches on full ImageNet. The model uses $h = 3$, $r = 1$ and is trained on CIFAR10. We use *e.g.* '400x' to denote an image is split into 400 patches. The reported BPD has standard deviation $\sim 0.02$ across multiple random seeds.

| Method | 1x | 16x | 25x | 100x | 400x |
|--------|------|------|------|------|------|
| BPD | 3.25 | 3.26 | 3.27 | 3.29 | 3.34 |

Table 6: Lossless Compression Comparisons. We compare against traditional images compression and neural network based models. For neural models, we report results where models are trained on CIFAR10 or ImageNet32 and tested on other ID or OOD test datasets. We use † to represent the best ID generalization and ⋆ to represent the best OOD generalization. (a) We down-sample CelebA to $32{\times}32$, see Appendix A.2. (b) The BPD $3.15$ reported in [48] is tested on $2000$ random samples from the full ImageNet testset, whereas we test HiLLoC on the whole testset with $49032$ images. The reported BPD of NeLLoC has standard deviation $\sim 0.02$ across multiple random seeds.

| Method | Size(MB) | CIFAR | SVHN | CelebA[a] | ImgNet32 | ImgNet64 | ImgNet |
|---|---|---|---|---|---|---|---|
| Generic | | | | | | | |
| PNG [6] | N/A | 5.87 | 5.68 | 6.62 | 6.58 | 5.71 | 5.12 |
| WebP [30] | N/A | 4.61 | 3.04 | 4.68 | 4.68 | 4.64 | 3.66 |
| JPEG2000 [39] | N/A | 5.56 | 4.10 | 5.70 | 5.60 | 5.10 | 3.74 |
| FLIF [46] | N/A | 4.19 | 2.93 | 4.44 | 4.52 | 4.19 | 3.51 |
| Train/test on one distribution | | ID | | | ID | ID | |
| LBB [18] | - | **3.12**$^{\dagger}$ | - | - | 3.88 | 3.70 | - |
| IDF++[3] | - | 3.26 | - | - | 4.12 | 4.81 | - |
| Trained on CIFAR | | ID | OOD | OOD | OOD | OOD | OOD |
| IDF [19] | 223.0 | 3.34 | - | - | 4.18 | 3.90 | - |
| Bit-Swap [23] | 44.7 | 3.78 | 2.55 | 3.82 | 5.37 | - | - |
| HiLLoC [48] | 156.4 | 3.32 | 2.29 | 3.54 | 4.89 | 4.46 | 3.42 |
| L3C [35] | 19.11 | 3.39 | 3.17 | 4.44 | 4.97 | 4.77 | 4.88 |
| **NeLLoC** ($r=0$) | **0.49** | 3.38 | 2.23 | 3.44 | 4.20 | 3.86 | 3.30 |
| **NeLLoC** ($r=1$) | 1.34 | 3.28 | 2.16 | 3.37 | 4.07 | 3.74 | 3.25 |
| **NeLLoC** ($r=3$) | 2.75 | 3.25 | **2.13**$^{\star}$ | **3.35**$^{\star}$ | **4.02**$^{\star}$ | **3.69**$^{\star}$ | **3.24**$^{\star}$ |
| Trained on ImgNet32 | | OOD | OOD | OOD | ID | OOD | OOD |
| IDF [19] | 223.0 | 3.60 | - | - | 4.18 | 3.94 | - |
| Bit-Swap [23] | 44.9 | 3.97 | 3.00 | 3.87 | 4.23 | - | - |
| HiLLoC [48] | 156.4 | 3.56 | 2.35 | 3.52 | 4.20 | 3.89 | **3.25**$^{\star\mathrm{b}}$ |
| L3C [35] | 19.11 | 4.34 | 3.21 | 4.27 | 4.55 | 4.30 | 4.34 |
| **NeLLoC** ($r=0$) | **0.49** | 3.64 | 2.38 | 3.54 | 3.93 | 3.63 | 3.37 |
| **NeLLoC** ($r=1$) | 1.34 | 3.56 | 2.26 | 3.47 | 3.85 | 3.55 | 3.31 |
| **NeLLoC** ($r=3$) | 2.75 | **3.51**$^{\star}$ | **2.21** | **3.43**$^{\star}$ | **3.82**$^{\dagger}$ | **3.53**$^{\star}$ | 3.29 |

## 5   Conclusion and Future Work

In this work, we propose the local autoregressive model to study OOD generalization in probabilistic image modelling and establish an intriguing connection between two diverse applications: OOD detection and lossless compression. We verify and then leveraged a hypothesis regarding local features and generalization ability in order to design approaches towards solving these tasks. Future work will look to study the generalization ability of probabilistic models in other domains *e.g.* text or audio, in order to widen the benefits of the proposed OOD detectors and lossless compressors.

**Acknowledgments**

We would like to thank James Townsend and Tom Bird for discussion on the HiLLoC experiments. We thank Ning Kang for discussion regarding the parallelizability of NeLLoC.

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
