# A Experiments Details

## A.1 Compute resources

All models were trained using an NVDIA GeForce RTX 2080 Ti and an NVDIA GeForce V100 GPU.

## A.2 Prepossessing of CelebA

We first perform a central crop with edge length 150px and then resize to $32 \times 32 \times 3$. We select the first 10000 images as our CelebA test set.

## A.3 Model Architecture

This section provides additional details concerning the model building blocks used in our experiments. All models share a similar PixelCNN [49] architecture, which contains a masked CNN as the first layer and multiple Residual blocks as subsequent layers, we next provide further details on both components.

**Masked CNN** structure is proposed in [49]. For our local model with dependency horizon $h$, one kernel of the CNN has size $k \times k$ with $k = 2 \times h + 1$. The masked CNN contains masks to zero out the input of the future pixels. There are two types of Masked CNN, which we refer to as mask A (zero out the current pixel) and mask B (allow connections from a color to itself), see [49] for further details. The first layer of our model utilizes mask A and the residual blocks use mask B.

**Residual Block** Each residual Block [49] contains the following structure. We use MaskedCNN$_B$ to denote a Masked CNN using mask B.

---
**Algorithm 1:** Residual Block
---
**Input**: $x_{input}$
$\quad h = \text{MaskedCNN}_B(x_{input})$
$\quad h = \text{ReLU}(h)$
$\quad h = \text{MaskedCNN}_B(h)$
$\quad h = \text{ReLU}(h)$
$\quad h = \text{MaskedCNN}_B(h)$
$\quad h = \text{ReLU}(h)$
**Return** : $x_{input} + h$

---

**Pixel CNN** Our full Pixel CNN and local pixel CNN shares the same backbone. The difference between the two models is that the full model has kernel size $3 \times 3$ for the second masked CNN layer in the Residual Block and, in contrast, the local model uses a kernel size of $1 \times 1$ for each of the masked CNN layers, in the Residual block. Crucially, this difference results in the receptive field of the full model increasing when stacking multiple Residual blocks whereas the receptive field of the local model does not increase. A Pixel CNN with $N$ residual blocks has the following structure:

---
**Algorithm 2:** Pixel CNN
---
**Input**: $x_{input}$
$\quad h = \text{MaskedCNN}_A(x_{input})$
$\quad h = \text{ReLU}(h)$
$\quad$**for** i from 1 to N:
$\quad\quad h = \text{ResBlock}_i(h)$
$\quad h = \text{MaskedCNN}_B(h)$
$\quad h = \text{ReLU}(h)$
$\quad h = \text{MaskedCNN}_B(h)$
**Return** : $h$

---

### A.3.1 OOD detection

**Gray images** For gray images, our full Pixel CNN model has five residual blocks and channel size 256, except the final layer which has 30 channels. The kernel size is 7 for the first Pixel CNN and the

kernel size is $[1\times1, 3\times3, 1\times1]$ for three masked CNNs in the residual blocks. Our local Pixel CNN model has one residual block and channel size 256, except the final layer which has 30 channels. The kernel size is 7 for the first Pixel CNN and the kernel size is $[1\times1, 1\times1, 1\times1]$ for three masked CNNs in the residual blocks. We use the discretized mixture of logistic distribution [41] with 10 mixture components. The models are trained using the Adam optimizer [20] with learning rate $3\times10^{-4}$ and batch size 100 for 100 epochs.

**Color images** For color images, our full Pixel CNN model has 10 residual blocks and channels size 256, except the final layer which has 100 channels. The kernel size is 9 for the first masked CNN and the kernel size is $[1\times1, 3\times3, 1\times1]$ for three masked CNNs in the residual blocks. Our local Pixel CNN model has 10 residual blocks and channels size 256, except for the final layer which has 100 channels. The kernel size is 7 for the first Pixel CNN and the kernel size is $[1\times1, 1\times1, 1\times1]$ for three masked CNNs in the residual blocks. We use the discretized mixture of logistic distribution [41] with 10 mixture components. The models are trained using the Adam optimizer [20] with learning rate $3\times10^{-4}$ and batch size 100 for 1000 epochs.

### A.3.2  Lossless compression

The NeLLoC is based on a Pixel CNN. Since it is a local model, the kernel size is $1\times1$ for all the kernels in the Residual block. Additional details regarding the first layer kernel size and number of residual blocks are found in the main text, Section 4. All models are trained using the Adam optimizer [20] with a learning rate of $3\times10^{-4}$ and batch size 100 for both the CIFAR dataset (1000 epochs) and the ImageNet32 dataset (400 epochs).

## B  Local Model

### B.1  Effect of Horizon Size for Color Images

In Section 2.2 of the main manuscript we show that, for a simple gray-scale dataset, the model can overfit to non-local features that are specific to the training distribution and thus degrade OOD generalization performance. For a more complex training distribution, *e.g.* CIFAR, we find that a model with limited capacity is less susceptible to overfit to non-local features. However, as observable in Table 7, when the local horizon size increases, the ID generalization continues to improve, whereas the OOD generalization remains stable. This is consistent with our hypothesis: local features are not shared between distributions and cannot significantly aid OOD generalization. We conjecture that, with the use of a more flexible base model *e.g.* PixelCNN++ [41], over-fitting to the non-local features will occur and thus result in familiar degradation of OOD generalization abilities.

Table 7: The generalization ability of local models with increasing horizon size. All models have residual block number $r{=}1$ and are trained on CIFAR10. The reported BPDs have standard deviation 0.02 across multiple random seeds.

| Method | h=2 | h=3 | h=4 | h=5 |
|---|---|---|---|---|
| (ID) CIFAR | 3.38 | 3.28 | 3.26 | 3.25 |
| (OOD) SVHN | 2.21 | 2.16 | 2.15 | 2.15 |
| (OOD) CelebA | 4.08 | 4.07 | 4.07 | 4.07 |

### B.2  Samples from Local Model

We show samples from a local model with $h{=}3$, trained on CIFAR $32\times32\times3$, in Figure 5. It can be observed in Figure 5a that the samples are locally consistent yet images do not possess much in the way of recognizable and meaningful global semantics. Figure 5b shows an example image of size $100\times100\times3$. This is made possible since the local model does not require sampled images to have a fixed size, *i.e.* size is not required to be consistent with the training data.

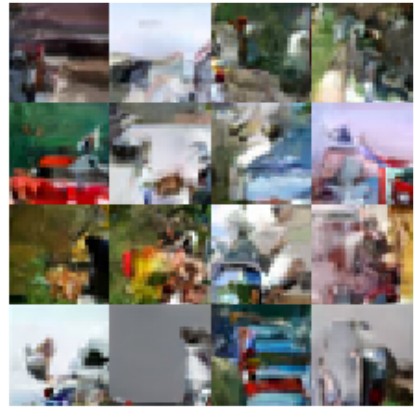 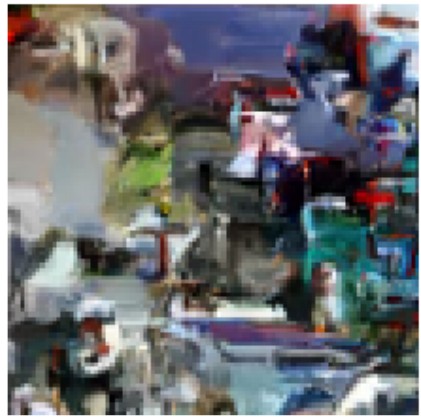

(a) 16 samples with size $32\times32\times3$   (b) One sample with size $100\times100\times3$

Figure 5: Samples from a local autoregressive model.

## C   Additional NeLLoC Experiments

We investigate NeLLoC under several different coders: Arithmetic Coding (AC) [50], Asymmetric Numeral System (ANS) [11] and interleaved-ANS [13] (iANS), in terms of both compression BPD and method run time, see Table 8 for details. We can find that ANS is slighter faster than AC yet sacrifices $\approx$0.01 BPD. The interleaved ANS allows compression (and decompression) of multiple data simultaneously, thus leads to a large improvement in the compression (decompression) time per-image. However, in comparison with AC, iANS sacrifices $\approx$0.03 BPD.

Table 8: Comparisons of NeLLoC with different coders. We implement NeLLoC with three different coders: Arithmetic Coding (AC), Asymmetric Numeral System (ANS), and interleaved ANS (iANS), and compare the compression performance in terms of BPD and compression time. Our test images are 200 samples from the CIFAR10 test set. Our models consist of three different structures with ResNet block numbers $r = \{0, 1, 3\}$ respectively. The initial three tables; 8a, 8b, 8c show results using a Macbook Air (M1, 2020, 8GB memory) CPU. We also report the results of NeLLoC-iANS with batch size 200 in 8d. Since large batch sizes require large memory, we conducted our experiments on a CPU Intel i9-9900k with 64GB of memory. Therefore, the results of 8d can not be directly compared with the results in the other three tables.

(a) Arithmetic Coding (AC)

| Res. num. | $r = 0$ | $r = 1$ | $r = 3$ |
|---|---|---|---|
| Test BPD | 3.35 | 3.25 | 3.22 |
| Comp. (s) | 0.87 | 1.34 | 2.16 |
| Decom. (s) | 0.95 | 1.42 | 2.24 |

(b) Asymmetric Numeral System (ANS)

| Res. num. | $r = 0$ | $r = 1$ | $r = 3$ |
|---|---|---|---|
| Test BPD | 3.37 | 3.26 | 3.23 |
| Comp. (s) | 0.82 | 1.29 | 2.06 |
| Decom. (s) | 0.83 | 1.30 | 2.07 |

(c) Interleaved ANS (batch size 10)

| Res. num. | $r = 0$ | $r = 1$ | $r = 3$ |
|---|---|---|---|
| Test BPD | 3.38 | 3.28 | 3.24 |
| Comp. (s) | 0.34 | 0.48 | 0.75 |
| Decom. (s) | 0.36 | 0.50 | 0.77 |

(d) Interleaved ANS (batch size 200)

| Res. num. | $r = 0$ | $r = 1$ | $r = 3$ |
|---|---|---|---|
| Test BPD | 3.38 | 3.28 | 3.24 |
| Comp*. (s) | 0.05 | 0.07 | 0.09 |
| Decom*. (s) | 0.07 | 0.08 | 0.11 |

# D Datasets

The CelebA [32], ImageNet [10] and SVHN [38] datasets are available for non-commercial research purposes only. The Fashion MNIST [51] and Omniglot [27] datasets are under MIT License. The KMNIST [9] is under CC BY-SA 4.0 license. We did not find licenses for CIFAR [25] and MNIST [28]. The CelebA dataset may contain personal identifications.

# E Societal Impacts

In this paper, we introduce a novel generative model and investigate related out-of-distribution detection, generalization questions. Local autoregressive image models, such as those introduced, may find application in many down-stream tasks involving visual data. Typical use cases include classification, detection and additional tasks involving extraction of semantically meaningful information from imagery or video. With this in mind, caution should remain prominent, when considering related technology, to avoid it being harnessed to enable dangerous or destructive ends. Identifcation or classification of people without their knowledge, towards control or criminilzation provides an obvious example.