# OpenReview forum: "On the Out-of-distribution Generalization of Probabilistic Image Modelling"
_NeurIPS.cc/2021/Conference — NeurIPS 2021 Poster_

### Official Review · Reviewer_t8Bq · 2021-07-12

**Rating:** 6
**Confidence:** 3

**Summary:**

The paper discussed the counterintuitive phenomenon in OOD detection using generative models, and proposed a new method called non-local feature density which fixed the issue and achieved the SOTA performance, because non-local features are shown to contain more semantic information. Further, the authors proposed an image compressor method, NeLLoC, based on the local features, because local features are shown to constitute a domain prior for all images so that NeLLoC is more generalizable to OOD images.

**Limitations And Societal Impact:**

The authors adequately addressed the limitations and potential negative societal impact of their work.

**Main Review:**

- Originality: the authors provides a new method for fixing the issue of higher likelihood of OOD inputs in the generative model based OOD detection methods. The general idea of the proposed method is similar to the previous methods: the full generative models include both local (non-semantic) and non-local features (semantic). Local image features appear largely common across the real-world image manifold and can be treated as a domain-prior;  only non-local features which typically contain semantic image information, are useful for OOD detection. So the designed OOD score is the unnormalized likelihood of non-local features, i.e. p(x|full model)/p(x|local model).
Compared with the existing likelihood ratio methods [35] and [38,39], the proposed method is more direct to address the issue of common local features dominating the likelihood. But the proposed method seems to be specific to the PixelCNN model architecture. I wonder if the method can be generalized to other generative models.

- Quality: the paper is well written, and the results are clearly presented.

- Clarity: It would be great if the algorithm for NeLLoC can be presented, describing the exact computational steps for compression or decompression.

- Significance: The proposed method is specifically designed based on the PixelCNN model. It would be great if the method can be generally applied to other types of generative models.

- Minor comments:
- Figure 3 (a) shows a slight overlap between the IND and OOD distributions, but Table 3 shows the AUROC for the same experiment is 1.00.
- Typo: eq(3), k->d


*** Update: Thank the authors for your responses. It is great that the framework can be applied to other autoregressive models such as Transformer. I wonder if Transformer model can achieve even better performance. After reading other reviews and responses, I decided to keep my score. ***

**Time Spent Reviewing:**

3

---

> ### Author Response · Authors · 2021-08-10
> **The framework can be applied to other autoregressive models**
>
> We thank the reviewer for the encouraging reviews!  We address the raised questions below.
> 1. As stated in line 80; our approach, based on an autoregressive structure, may also be implemented by e.g. transformer or PixelRNN models. We think our proposed method in the current form will not easily generalize to a latent variable model (e.g. VAE) or change of variable model (like flow) since such models directly model all the information in the data. However, the hierarchy latent variable may be possible to learn local features in the shallow layers and global features in deep layers, we leave that for future exploration.
> 2.  We provide source code for NeLLoC by following the anonymous link: https://anonymous.4open.science/r/NeLLoC-AC-B4FE. Our code provides clarifications for the implemented compression and decompression steps. Post-publication, we will provide fully de-anonymized and documented source code towards ease of reproducibility.
> 3. As stated in the Table 3 caption; reported values are rounded to three decimal places, we, therefore, plot 1.000 when the AUROC is larger than 0.9995.
> 4. We thank the reviewer for flagging typos and correct these appropriately.

---

### Official Review · Reviewer_XCL4 · 2021-07-13

**Rating:** 5
**Confidence:** 4

**Summary:**

Summary: This paper study on applying generative models on unsupervised out-of-distribution detection. Authors hypothesize that local features common to both in-distribution and out-of-distribution data dominate the likelihood of generative models. The authors show evidence of the hypothesis via comparing the fully autoregressive model’s likelihood and autoregressive model with local dependencies. Motivated by the results, the authors propose a ratio between the likelihood of model trained with full-dependencies and the likelihood of model trained with local dependencies. The authors show that the proposed metric outperforms conventional generative-model-based methods on CIFAR-10 and FashionMNIST domains. Finally, the authors apply local autoregressive models on universal lossless compression. The proposed compression scheme outperforms conventional methods.


**Ethical Concerns:**

I could not find any ethical issues on the paper.

**Limitations And Societal Impact:**

While authors did not focus on explaining the potential negative social impact of their work, I believe that is not such an issue.

**Main Review:**

UPDATE :
I acknowledge that I have read the author response as well as the other reviews.
Although authors replied my concerns on the novelty, I still believe the main intuition of the paper, "directly model the in-distribution dataset, using only local feature information" is straighforward application from the idea of previous works.
Due to the previous reasoning, I insist deleting section 3.2 and Figure 1.
However, I understand the virtue of the method on unsupervised novelty detection, which doesn't need specific augmentations.
Furthermore, I appreciate the clarifications on lossless compression.
Therefore, I am changing my score to 5 and confidence to 4.

=========================

UPDATE (2):

While this is relatively minor, I question the following points in the reply.

Our major goal of OOD detection experimental work is to show that our simple and principled approach can perform comparatively strongly. We believe our experimental results strongly support this claim. At the time of submission, we are not aware of any data augmentation free, model ensemble free methods, capable of consistently outperforming our approach.

=> Consider looking at the paper "Multiscale score matching for out of distribution detection" [https://arxiv.org/pdf/2010.13132.pdf]

While the experiments are performed only on CIFAR-10 vs SVHN, the stark contrast on the performance reasons me whether the scheme proposed by the authors is competitive, or just a method that can be outperformed by a better generative model.
===========================================

Update (3)

I acknowledge that I read the response from the authors as well as the other reviews. I believe that the methodology proposed in the paper shows State-of-the-art performance among augmentation-free unsupervised out-of-distribution detection methods. However, I'm still on the side that the main novelty of the paper is relatively minor. Therefore, I still stick to the current ratings.

==========================================

Pros
-       Application on the universal lossless compression task is interesting, while this is not the first work to apply generative models on lossless compression. Furthermore, the proposed method outperforms conventional methods
Cons

-       My deepest concern on the paper is that the proposed hypothesis is not new and published in NeurIPS 2020. We can find in [1] where the authors verified that GLOW and RealNVP encode representations overfitted on local pixel correlations. I would not be surprised that the same thing happens to PixelCNN since they all failed in unsupervised anomaly detection [2] and they both exhibit an encoder-decoder-like structure. As a result, this paper sounds like an extension of existing ideas to a different generative model.

-       Experiment results are limited. In table 3, the authors compare their proposed metric against diverse OOD detection methods. However, the authors compared their results against comparably weak baselines. I can find other published practices [3] [4] in unsupervised novelty detection that outperform the result.

Overall, I think the main idea of this paper is nothing different from the idea of [1], and the experimental results are not compelling or convincing compared to recently published unsupervised novelty detection methods [4]. Therefore, I am leaning towards rejecting the paper.

Comments

-       Authors should cite [1].

-       I think FashionMNIST is comparably too “easy” dataset for verifying out-of-distribution detection performance. I suggest experimenting on diverse in-distribution datasets, e.g., TinyImageNet, CelebA, LSUN. Furthermore, there are too many blanks in the experiment result that it is hard to convince that the proposed metric improves over conventional methods. Furthermore, it would be beneficial to compare the performance of the paper against more diverse unsupervised OOD detection methods.

-       I would like to see the effect of variation in local horizontal length(l) and kernel size(k) as means of OOD detection performance across different in-distribution datasets. It would be beneficial to show whether the optimized hyperparameters can be applied in other unobserved datasets.


References.
[1] Why normalizing flows fail to detect out-of-distribution data. Polina Kirichenko∗ , Pavel Izmailov∗ , Andrew Gordon Wilson

[2] Do deep generative models know what they don’t know? Eric Nalisnick et al.

[3] Novelty detection via blurring. Sungik Choi and Sae-Young Chung

[4] CSI: Novelty Detection via Contrastive Learning on Distributionally Shifted Instance. Jihoon tack, Sangwoo mo, Jongheon Jeong, and Jinwoo Shin.


**Time Spent Reviewing:**

4

---

> ### Author Response · Authors · 2021-08-10
> **Contributions and significance of the results**
>
> We thank the reviewer for their time and valuable feedback that improves the quality of our work. We are encouraged by the positive comments relating to our method performance and also the interest in application to compression tasks.
>
>
> **Connection to prior work [1].**
> We appreciate the pointer to relevant work cite this paper in our updated version, appropriately. In contrast to the highlighted paper, our contributions can be summarized as follows:
>
> 1. We attribute the low-level feature hypothesis appropriately (we cite Schirrmeister et al. on line 68, which is very similar to [1]). We then go further; we are the first work to build an explicit model in order to verify the phenomenon that 'the likelihood is dominated by local features'.
> 2. Specifically, we provide direct evidence in support of the hypothesis by observing the BPD in both local and full models. Our work goes beyond simply testing correlations between local features or visualizing intermediate network layers.
> 3. The highlighted alternative work provides an explanation for why Flow fails, yet stops short of proposing an OOD detection strategy and further, does not evaluate AUROC over dataset pairs.
> 4. The proposed OOD approach is (i) simple; it does not require augmentation or validation data and (ii) principled; the score has a likelihood interpretation and may be considered an un-normalized non-local likelihood.
> 5. We propose the first method that employs a product of experts to model entire images for the OOD detection task.
> 6. Our local (PixelCNN) model is based on an auto-regressive structure that significantly differs from an encoder-decoder structure. Our model contains no encoder or decoder (c.f. paper [1]).
>
> **Results are limited, additional datasets.**
>
> 1. We note that the two papers [3], [4] that the reviewer mentions both necessitate data augmentation. Specifically [3] requires construction of blurry images. Alternatively, the design of data transformations, to facilitate contrastive learning, is necessary in [4]. Data augmentation may improve performance in some cases, however, it heavily depends on manual heuristics and design of data transformations. In stark contrast, our approach requires no data augmentation whatsoever. We thank the reviewer for the pointers and update our manuscript by adding these works to our comparisons, highlighting this key difference.
>
> 2.  Greyscale datatsets (FashionMNIST/MNIST) do not constitute an easier OOD detection task in comparison to data containing color images. It is well understood that the distributions of such data support a more discrete topology and much lower dimensional manifolds. Therefore, methods that perform well on color images may certainly underperform on greyscale data. We refer the reviewer to the WAIC method in Table 3 of our paper in support of this point.
>
> 3.  We actively select not to include dataset pairs such as CIFAR10/ImageNet or CIFAR10/CIFAR100, since these data contain duplicate classes (e.g. horse or ship) and thus cannot be treated as two disjoint datasets (or OOD dataset), see the discussion in Appendix A of [5].  We thus alternatively choose to conduct experiments on datasets where humans are able to reliably distinguish OOD data.  This is also the reason why recently published work (ICML 2021) [6] still only considers Fashion/MNIST and CIFAR/SVHN datasets pairs. We realize there are many bars in the KMNIST section, we will reform the table to present a neater comparison.
>
> 4. Our major goal of OOD detection experimental work is to show that our simple and principled approach can perform comparatively strongly. We believe our experimental results strongly support this claim. At the time of submission, we are not aware of any data augmentation free, model ensemble free methods, capable of consistently outperforming our approach.
>
> We would appreciate it if the reviewer could take the time to assess our clarifications, additional details and indicate whether this addresses your concerns and changes your assessment of our work. Please feel free to point out anything else you might consider relevant.
>
> **Local horizon size**
>
> We thank the reviewer for this suggestion. We only conducted the ablation study in the context of the lossless compression application, see Table 5. We will update our manuscript to add the suggested ablation study in our revised version. We emphasize that the only difference between the local and full models is the horizon length. When the local horizon length increases, the local model converges to a full model. In such cases, the ratio becomes one everywhere and OOD detection will fail. This phenomenon suggests the importance of using a local model.
>
> **Other contributions**
> We highlight that our additional contributions are also significant, e.g.
> 1. Establishing a boundary-breaking connection between two different applications by exploring the model generalization. See related reply to reviewer t6SW.
> 2. We use knowledge, gained from OOD detection, to inspire a new lossless compression algorithm that achieves state-of-the-art generalization with low computational requirements. See the runtime performance section at the start of our rebuttal.
>
> ## Additional reference, [1]-[4] are  in the reviewer's feedback
> [5] Serrà, Joan, et al. "Input complexity and out-of-distribution detection with likelihood-based generative models." ICLR 2020.
>
> [6] Havtorn, Jakob D., et al. "Hierarchical VAEs Know What They Don't Know." ICML 2021.

---

> ### Author Response · Authors · 2021-08-18
> **Reply to the updates**
>
> We thank the reviewer for their time to partake in the discussion and for further considering our replies and additional points we highlight.
>
> ### Novelty of contributions in OOD detection
>
> We would respectfully disagree with the suggestion to remove Figure 1 and section 3.2 for the following reasons:
>
> 1. In line 50 we clearly state that the phenomenon: ``OOD data may have a higher likelihood than in-distribution data, for a probabilistic model'' is observed in previous work [1], and that it is not our contribution. We would however argue that the figure in question helps to make our work self-contained and increases understanding for readers who are less familiar with OOD detection.
> 2.  In line 68, we cite the paper that initially explicitly posed the hypothesis: ``likelihood is dominated by local features''. We thank the reviewer for drawing [3] to our attention, where the hypothesis is implicitly raised in relation to flow models. We add this citation to our work.
>
> Crucially, we do not attempt to argue that the stated hypothesis is part of our contribution. Alternatively, we emphasize that our contribution, stated in section 3.2, proposes a novel path to verifying, and collecting evidence in support of, such hypotheses. We evaluate the local likelihood using a local model such that the dominant effects can be directly observed, for the first time, in terms of the likelihood (BPD). In comparison to previous verification attempts [2-3], our method is more principled (using pure likelihood comprisons).
>
> This verification method futhur enables our related contributions:
>  1. An OOD based method with a principled likelihood interpretation.
>  2. A lossless compressor with improved generalization.
>  3. An illustration of the unconventional connection between OOD detection and lossless compression.
>
> We further emphasize that our model of the data is a ***product-of-experts model*** (Equation 5), and yet critically one expert is locally autoregressive. The score used for OOD detection is the (unnormalized) likelihood of a non-local (semantic) model. We are unaware of any such product-of-experts style approach being used previously in the OOD detection field.
>
> These points lead us to disagree with the notion that our OOD detection method constitutes a simple extension of any previous method.
>
> ### Additional analysis relating to [4]
>
> We thank the reviewer for highlighting an additional relevant paper [4] (ICLR 2021). We note that this was published very near to the NeurIPS submission deadline (May of 2021), however, we agree the paper is pertinent and we will now include a citation in our updated work. Towards initial comparison with [4], we here compare their results with our own. Their work actually reports results for both SVHN vs CIFAR (their Table 2) and FashionMNIST vs MNIST (their Table 14, in Appendix A.6.).
>
>
> | methods/datasets          | SVHN vs CIFAR  | FashionMNIST vs MNIST|
> | ------------- | -----:|-----:|
> |    MSMA GMM      |  91.9  | 82.56|
> |MSMA Flow        |  93.4 |82.05 |
> |MSMA KD Tree |99.1| 69.32|
> |Ours| 96.9| 100|
>
> In the work of [4], three variants of their MSMA method are proposed. We observe that our method is competitive in the SVHN vs CIFAR case (lower than their MSMA KD Tree, and improves upon the other two approaches). Further, we note our approach offers consistent and significant improvements for FashionMNIST vs MNIST. Our method is consistently reliable and we would suggest is also simple and principled in comparison to such two-stage approaches. We would ask the reviewer to comment further on these observations.
>
> Further; the results of [4] further support our previous rebuttal argument (see point 2 in "***Results are limited, additional datasets***"); grey-scale images are typically -not- easier than color images for the OOD detection task.
>
> ### Others
> Finally, beyond OOD detection, we also believe our lossless compression contribution to be significant. Our reported observations may influence and open one direction of AI-based compression research towards the modeling of local images (cf. global).  See e.g. our updated 'vectorized rANS implementation of NeLLoC' rebuttal discussion.
>
> We thank the reviewer for their extended time and ask them to further take into consideration the additional points highlighted here when providing their assessment of our work.
>
> ### Reference
> [1] Do deep generative models know what they don’t know? Eric Nalisnick et al.
>
> [2] Understanding anomaly detection with deep invertible networks through hierarchies of distributions and features. Schirrmeister, Robin Tibor, et al.
>
> [3] Why normalizing flows fail to detect out-of-distribution data. Kirichenko et al.
>
> [4] Multiscale Score Matching for Out-of-Distribution Detection. Mahmood, Ahsan et al.

---

> ### Author Response · Authors · 2021-09-01
> **Thanks for the reviews**
>
> We want to thank the reviewer for the comments and questions.
>
> We are wondering have our replies addressed your concerns or make you furthur change the score of the paper? If there are any additional questions, we can try our best to solve them in the last minute. We're very much looking forward to hearing your further feedback on the response.
>
> Best!
>
> Authors

---

> ### Author Response · Authors · 2021-09-02
> **Novelty of our paper in the OOD detection**
>
> We thank the reviewer for their frequent communication and discussion, it is appreciated and helpful. We are encouraged by the acknowledgment of our lossless compression contributions. Towards addressing the final remaining novelty concerns, regarding our OOD contributions, we highlight and restate the significance of these parts at the start of the rebuttal and would draw the reviewer's attention to this summary for their consideration.

---

### Official Review · Reviewer_t6SW · 2021-07-15

**Rating:** 6
**Confidence:** 3

**Summary:**

This paper proposes a new probabilistic image modeling approach based on a local autoregressive model. The local autoregressive model is implemented like the masked convolution operation in the pixelCNN model. The model is applied to two downstream tasks of OOD detection and lossless image compression. The OOD score is computed using the ratio between the likelihood from the full model versus the one from the local model. The idea is simple but shows some improvement over previous models.



**Limitations And Societal Impact:**

There is no explicit discussion on the limitations or societal impact of the proposed work, but there appears to be no direct negative societal impact from this work. It will be helpful if more discussion on the limitations of the proposed method is included.

**Main Review:**

[Update] I have read the author responses and comments by the other reviewers. My previous concerns about clarity and significance have been partly resolved, so I'm changing my score from 5 to 6.
---------------------------------------------------------------------

The main idea comes from the intuition that the local features are shared between different image datasets and dominate the generalization ability, which is interesting.

In terms of clarity, it was quite hard to follow the main idea of the paper at first, probably because it is not clearly introduced how the core model is connected to the two downstream tasks which seem quite unrelated. Moreover, the target tasks are explained in sections 2, 4, and 5 while the main component of the proposed approach is explained in sec 3. Also, the methods and results are mixed in the same section.  For better readability, the overall structure needs to be improved, for example, the authors may provide more introduction about the overall flow of the paper in the beginning and rearrange the following materials.

It is not clear from Table 3  if the performance gain over the previous methods for OOD detection is significant. Actually, it is not clear where the main focus of this paper is.  It might be better to choose one target task (probably OOD detection) and put more emphasis on that by investigating the performance behavior in more details.

As the author argues, time complexity is an important factor in the case of lossless compression.  I wonder how the actual encoding and decoding time compares with those of other methods.

**Time Spent Reviewing:**

6

---

> ### Author Response · Authors · 2021-08-10
> **Relationship between applications and paper structure**
>
> We thank the reviewer for their time and valuable feedback that improves the quality of our work. We are encouraged by the positive comment regarding our underlying main idea.
>
> **Relationship between applications and paper structure:**
>
> We agree with the reviewer that the relation between tasks and the core model may not be immediately obvious. Our paper aims to demonstrate an elegant affinity between the considered tasks and the resulting model design. We update our manuscript towards making this connection more clear. To further address raised clarity concerns we re-emphasize hereafter the structure of the paper:
>
> 1. We begin by defining OOD generalization in terms of probabilistic models.
> 2. We highlight that lossless compression requires *encouragement* of generalization and, conversely, that the OOD detection task requires *discouragement* of the generalization. We thus build a connection between the considered tasks.
> 3. We verify the assumption that local features are shared between different datasets and dominate the generalization. From this, we can proceed to intuitively leverage the previously noted task connection by:
>     - Exclude local features for the task of OOD detection, due to their noted generalization discouragement
>     - Include exclusively local features for lossless compression, which encourage generalization.
> 4. In practice, we build a model that only models the local features;
>     - Divides the full model likelihood by the local model likelihood for the OOD detection task (equivalent to an un-normalized likelihood of a non-local or semantic model).
>     - Only uses the local model for the lossless compression task.
>
> 4.  Finally, we report that the proposed method achieves competitive results for both applications.
>
>
> We hope that this analysis addresses your concern in two ways. Firstly we consider an illustration of the fundamental connection between these two tasks to be an important motivation for our work and utilize the findings from one task to significantly benefit the other. We believe such a boundary-breaking connection to be both of interest to the community and possesses the ability to benefit both research areas. Secondly, our investigation adds novelty; such an analysis of the affinity between these tasks has, to the best of our knowledge, never been conducted and we demonstrate the efficacy of leveraging this relationship.
>
> We would appreciate if the reviewer could take the time to assess our additional explanations and indicate whether this addresses the raised concerns and changes the assessment of our work. Please feel free to point out anything else that might be considered relevant.
>
> **Significance of Table 3**
> Comparing to the existing OOD detection methods, our approach is (i) simple; it does not require augmentation or validation data and (ii) principled; the score has a likelihood interpretation and may be considered an un-normalized non-local likelihood. We believe the significance is in the elegance of the proposed framework.
>
> Therefore, our major goal of providing the evidence found in Table 3 is to show that our simple and principled approach can perform comparatively strongly. At the time of submission, we are not aware of any data augmentation free, model ensemble free methods, capable of outperforming our approach.
>
> **Encoding and decoding time**
> Please see our runtime performance message at the beginning of the rebuttal.

---

### Official Review · Reviewer_kxNK · 2021-07-16

**Rating:** 6
**Confidence:** 3

**Summary:**

A surprising observation in the OOD literature is that OOD detection based on density estimation fails on some image datasets since the learned density tends to be higher on OOD data. A hypothesis for this phenomenon is that the likelihood is dominated by the local features of the image, resulting in the high-level features being ignored. Motivated by this observation, this paper proposes a model that learns local features of images. This model is used for two different applications: (1) by combining it with a full density estimation model, it gives a better OOD detection method, achieving state-of-the art performance (2) by pairing the model with the artihmetic coding algorithm, it gives a dataset compression method that achieves state-of-the-art compression rates on in- or out-of-distribution datasets.

**Limitations And Societal Impact:**

I cannot think of limitations beyond what is mentioned in the paper.

**Main Review:**

Both the ideas of utilizing the problematic and puzzling phenomenon of failures of density-based OOD detectors for a different approach (compression) and likelihood-ratio of the full and local model are novel and valuable. The reported results are impressive, making this method a promising approach for both these tasks. In the compression task, being able to achieve such results with way smaller models is also impressive.

Questions/comments:

- I am wondering how the runtime of different methods compare. The paper does comment on how the runtime of their proposed method scales and regarding its parallelizability, but an additional table of results showing the exact time for compression/decompression is useful.

- Why are many entries in table 3 missing? (although, they are irrelevant for all but the last column, since the AUROC of the proposed method 1)

---
Post-author response update: I appreciate authors taking the time to respond and provide table of run-times. I have read other reviews and responses and decided to keep my score the same.

**Time Spent Reviewing:**

7

---

> ### Author Response · Authors · 2021-08-10
> **Runtime performance**
>
> We thank the reviewer for their time and valuable feedback that improves the quality of our work. We are encouraged by the positive comments regarding our novelty and impressive result quality.
>
> 1. We believe the runtime performance investigation, located at the start of our rebuttal, can help to address the reviewer's concern.
>
> 2. We note that many works do not consider FashionMNIST and KMNIST pairs. This unfortunately results in a number of unavoidable blank Table column entries. We further remark that other recent OOD detection papers, e.g. published at ICML 2021 [1], only consider two experiments (FashionMNIST/MNIST and CIFAR/SVHN pairs). We consider it impractical to run all competing methods; some code is not public and as such it becomes difficult to reproduce methods faithfully. We consider our results to be significant and we offer a meaningful and direct comparison with four other methods.
>
> ## Reference
> [1] Havtorn, Jakob D., et al. "Hierarchical VAEs Know What They Don't Know." ICML 2021.

---

### Author Response · Authors · 2021-08-10
**Running time performance (see another message for an improved implementation)**

# Running time performance
Multiple reviewers raise questions regarding the method runtimes. We highlight that compression and decompression speed is dominated by implementation language choice (eg. C++ is much faster than python), coder choice (rANS is faster than AC), and other engineering tricks. We actively select to only compare computational complexity in our manuscript, which we consider a fair comparison of different models.

In the following section, we give a demonstration of computational performance c.f. another VAE based compression method: HiLLoC. We briefly highlight key differences here:
- NeLLoC:  PyTorch implementation with decimal AC coder, the encoding and decoding is autoregressive;
- HiLLoC [1]: JAX implementation with rANS coder, the encoding and decoding is fully parallelized.

We note that comparisons are not entirely fair, yet duly demonstrate the competetive performance of our method.

## Demostration

The code of all the comparisons can be found at this anonymous link: https://anonymous.4open.science/r/NeLLoC-AC-B4FE .

The model structure of NeLLoC is a local PixelCNN with one CNN layer followed by two 1x1 conv nets (0 ResNet). We use a discretized logistic-uniform mixture distribution as emission distribution. The model size is only *105 KB* (size of PyTorch .pt file).

### Performance of NeLLoC on CPU (MacBook Air 2020, M1 chip)
Here we list compression performance. All experiments are tested using a single CPU (M1 chip, MacBook Air 2020). See demo.ipynb file for details.

| NeLLoC   (105 KB)     |    SVHN        | CIFAR  |
| ------------- |:-------------:| -----:|
| BPD      | 2.38 | 3.64 |
| Compression time (s)     | 0.376      |   0.434 |
| Decompression time (s)      | 0.401      |  0.471 |

### Comparison to HiLLoCon CPU (Razor Blade 15, i7-10750H)
We do not manage to run HiLLoC (JAX implementation) on the MacBook Air M1 chip. Alternatively, we run both models on a Razor Blade 15 with CPU i7-10750H. We test two models for compressing a single image (see the file hilloc_comparisons for details).


| Method |    BPD  |loading time (s) | Compression time (s)| Decompression time (s) | CPU memory usage |
| ------------- |:-------------:| -----:| -----:|-----:|-----:|
| NeLLoC      | 3.99 | 0.0013| 0.71| 0.82|  0.126 MB|
| HiLLoC     | 4.0  |0.682    |   5.21 | 0.35 |  317.0 MB |

We observe that NeLLoC (with decimal arithmetic coding) is slightly slower than HiLLoC (with rANS) in decompression, but much faster for compression and loading time. We also find the CPU memory cost is negligible to run NeLLoC. This entails that our approach is much less demanding for the device and therefore more applicable in practice. We also noticed that the NeLLoC can be further accelerated by using a faster coder (like integer AC or rANS), we leave that for future work.

## Significance

A major criticism of deep generative-based compression methods is that they need huge computational resources (e.g. modern GPU), which restricts the practicability of such methods (comparing to, e.g.  PNG or FLIF). However, we show that our method can easily run on a CPU with negligible CPU memory cost (so massive images can be parallelized for compression or decompression, we leave to the future work). This provides evidence, for the first time, that computation need not be a limiting factor for learning-based lossless compression. We believe with a better implementation (C++, a faster coder like rANS, or other engineering improvements), NeLLoC represents a powerful candidate towards the replacement of traditional image compression methods.

Additionally, our paper shows that by only modelling local features, NeLLoC can exhibit better generalization ability in comparison with other global generative model-based methods. This suggests that if the goal is universal-purpose lossless compression, we should focus on local models rather than traditional global generative models.

## Reference
[1] Townsend, James, et al. "Hilloc: Lossless image compression with hierarchical latent variable models." arXiv preprint arXiv:1912.09953 (2019).

---

### Author Response · Authors · 2021-08-17
**A vectorized rANS implementation of NeLLoC**

We provide here a vectorized rANS [1] python implementation of NeLLoC  (the previously provided runtime performance of NeLLoC is based on a decimal AC coder).
The code can be found: https://anonymous.4open.science/r/NeLLoC-batch-1F60/README.md. Other details are the same as stated in the "Running time performance" message.

### Results and significance
The results are produced on a single CPU (Razor Blade 15, i7-10750H), the time is averaged over 1000 CIFAR images.

| NeLLoC   (105 KB)           | CIFAR  |
| ------------- | -----:|
| Compression time (s)          |   0.014 |
| Decompression time (s)         |  0.035 |

We can see the speed is 50x faster in compression and 20x faster in decompression comparing to the decimal AC implementation, see the table in the "Running time performance" message. Therefore, we show that the NeLLoC can not only achieve competitive BPD comparing to the deep generative model-based compression, but also eliminates the computation bottleneck of the AI-based lossless compression models.  We believe it is a significant contribution to the lossless image compression application.

### Other contributions of our paper in addition to lossless compression.
Despite the success in lossless compression, we want to re-emphasize other contributions in our paper which we believe are also very important both in theory and practice.
1. Build a boundary-breaking connection between OOD detection and lossless compression by exploring the generalization of the probabilistic models. See the reply to reviewer t6SW for details.
2. Verify the assumption that the local feature dominates the generalization by directly building a local model. We believe such a direct-modeling approach is more elegant and convincible comparing to previous methods.
3. Provide an elegant OOD detection framework that using a novel product of the local-global experts model, and allows a principled likelihood interpretation of the score function. This elegant framework also achieves competitive performance without any data augmentation. See the reply to reviewer XCL4 for details.

We would appreciate it if the reviewers could take the time to assess our additional explanations and indicate whether this addresses the raised concerns and changes the assessment of our work. Please feel free to point out anything else that might be considered relevant.

[1] Duda, Jarek. "Asymmetric numeral systems." arXiv preprint arXiv:0902.0271 (2009).

---

### Author Response · Authors · 2021-09-02
**Novelty and contributions in the OOD detection**

We are encouraged by the fact that all reviewers acknowledge our ***lossless compression contributions***. We here highlight our additional contributions and novelty in the task of OOD detection to further address the comments of reviewer XCL4.

1. We propose a novel path to verifying the recently proposed hypotheses: ``likelihood is dominated by local features''. We evaluate the local likelihood using a local model such that the dominant effects can be directly observed, for the first time, in terms of the likelihood (BPD). In comparison to previous verification attempts [1-2], our method offers a principled approach (likelihood comparison).

2. Based on our verification, we propose a new model for OOD detection -- a product of experts that enables OOD detection scoring and additionally allows for a likelihood interpretation: the likelihood of a semantic (non-local) model. We observe that our principled method achieves state-of-the-art results. We are not aware of any other OOD detection methods using a product of experts model or that afford a score possessing a semantic likelihood interpretation.

3.  We proposed the local auto-regressive model as a building block (the local expert) of the product-of-experts model. The proposed local model has been further applied to the lossless compression task.

4. We illustrate an unconventional connection between OOD detection and lossless compression by using the concept of generalization. We believe the connection we highlight to be of both interest and value to the community and has the potential to inspire further work towards linking these largely independent research fields.

In summary, our contribution to the OOD detection task is both novel and significant, enabled by our simple and yet principled methodology.

### Reference
[1] Understanding anomaly detection with deep invertible networks through hierarchies of distributions and features. Schirrmeister, Robin Tibor, et al.

[2] Why normalizing flows fail to detect out-of-distribution data. Kirichenko et al.

---

### Decision · Program_Chairs · 2021-09-27

**Decision:**

Accept (Poster)

**Comment:**

The main concerns about this work shared by the reviewers were around novelty and presentation. One reviewer felt that a key idea underlying the work has already featured in several other works in recent years, and thus the novelty of the work is limited. I am inclined to agree with this, however, I believe this work could be a useful reference to help more people understand the intricacies of OOD detection and the biases of likelihood-based models (especially autoregressive models), and the practical demonstration that a tiny local model is all you need for efficient lossless image compression, is valuable in itself.

Therefore, I have decided to recommend acceptance. This is of course conditional on the authors including in the manuscript all the additional results they have provided, and reworking the parts that were deemed unclear as set out in the authors' responses.